REGISTERED REPORT PROTOCOL

# Comparing phonological and orthographic networks: A multiplex analysis

**Pablo Lara-Martínez[1], Bibiana Obregón-Quintana[1], Cesar F. Reyes-Manzano[2], Irene López-Rodríguez[2], Lev Guzmán-Vargas[2] \***

**1** Facultad de Ciencias, Universidad Nacional Autónoma de México, Ciudad de México, México, **2** Unidad Profesional Interdisciplinaria en Ingeniería y Tecnologías Avanzadas, Instituto Politécnico Nacional, Ciudad de México, México

\* lguzmanv@ipn.mx

## Abstract

The complexity of natural language can be explored by means of multiplex analyses at different scales, from single words to groups of words or sentence levels. Here, we plan to investigate a multiplex word-level network, which comprises an orthographic and a phonological network defined in terms of distance similarity. We systematically compare basic structural network properties to determine similarities and differences between them, as well as their combination in a multiplex configuration. As a natural extension of our work, we plan to evaluate the preservation of the structural network properties and information-based quantities from the following perspectives: (i) presence of similarities across 12 natural languages from 4 linguistic families (Romance, Germanic, Slavic and Uralic), (ii) increase of the size of the number of words (corpus) from $10^4$ to $50 \times 10^3$, and (iii) robustness of the networks. Our preliminary findings reinforce the idea of common organizational properties among natural languages. Once concluded, will contribute to the characterization of similarities and differences in the orthographic and phonological perspectives of language networks at a word-level.

## Introduction

Many studies focused on the complexity of natural language have pointed out that language is the manifestation of different levels of complex organization [1–4], ranging from semantics [5] to syntax [6, 7] or even emotional components [8]. Of particular interest are the applications of network science on language organization, where these levels of complexity may be explored by means of single [9, 10] and multilayer graphs [11, 12]. A number of studies have reported emergent organizational properties in language based on associations of semantics, orthographic similarities [13] and phonetics [14, 15]. In many of these networks, the behavior of connectivities -the number of neighbors of a given node- is found to follow a distribution with a tail, which can be short or large. For instance, Arbesman et al. [16] reported that for phonological networks, the degree distribution can be well described by a truncated power law for several languages. For orthographic networks, Trautwein et al. [13]

**Data Availability Statement:** All corpora used in this study are available from the https://doi.org/10.6084/m9.figshare.12735380.v4 database.

**Funding:** This work was partially supported by programs EDI and COFAA from Instituto Politécnico Nacional and Consejo Nacional de Ciencia y Tenología, México. No additional external funding was received for this study. The funders had no role in study design, data collection and analysis, decision to publish, or preparation of the manuscript.

**Competing interests:** The authors have declared that no competing interests exist.

described that the distribution of connectivities for mental lexicon of students at elementary level, has a power law tail and the network exhibits a small-word property. Despite the variety of characterizations of language from the network's perspective, only a limited number of studies have incorporated the multi-layer aspects of language. Here, we consider a bi-layer approach of the analysis of orthographic and phonological language networks. Our procedure is based on the mapping of words into a two-layer network where nodes are words, and where connections are defined if an appropriate distance similarity is considered. In general, distance similarity between two strings, A and B, can be defined as the minimum number of edit operations needed to transform A into B. In our study we will consider the Damerau-Levenshtein (DL) as a proxy of the similarity between two words. It is recognized that, for many natural languages, there is not a biunivocal correspondence between how a word is spelled and its corresponding pronunciation, for instance, there is not a biunivocal correspondence between graphemes and phonemes. In fact, it is more likely to be observed in particular situations like homography (when a letter corresponds to two phonemes), digraphy (two letters correspond to one phoneme or viceversa), heterography (one phoneme corresponds to two or more letters), etc.

When comparing orthographical and phonological networks, an important question would be if the local and global connectivity patterns exhibit similarities. As well as what kind of differences can be identified, more specifically, in the context of psycholinguistics studies. The latter suggesting that the acting mechanisms on the cognitive processes, such as word recognition and retrieval, are particularly different than the orthographic organization.

## Proposed hypothesis and research plan

Our study is based on the premises that network representation of both syntax and phonological networks capture the most representative features of each network. In this sense, different questions can be asked. Our study focuses on the following three research questions:

- What are the characteristics of multiplex orthographic-phonological language networks?

- Would the connectivity patterns from orthographic and phonological networks reveal similarities and differences between them?

- How does orthographic structure varies in relation to phonological patterns across several natural languages?

There is enough evidence that phonological and grammatical networks exhibit common properties and differences. We shall focus on the evaluation of properties both locally and globally to show the differences between each layer while quantifying them at a bi-layer network (multiplex). To strengthen our study we initially intend to carry out the analysis in four natural languages (Spanish, English, German and Russian) via a $10^4$ word corpus. The plan for a secondary stage contemplates two considerations: (i) increase the corpus size from $10^4$ to $50 \times 10^3$ words and (ii) expand the analysis to 12 languages belonging to 4 different linguistic families (Germanic, Romance, Slavic and Uralic).

## Data analytic and proposed analyses

### Methods

The study of complex networks has incorporated the analysis of systems, for which, multiplex modelling is more suitable. In these cases nodes are located in layers with connections among them and the nodes are common to all layer-networks. A number of real-world and simulated

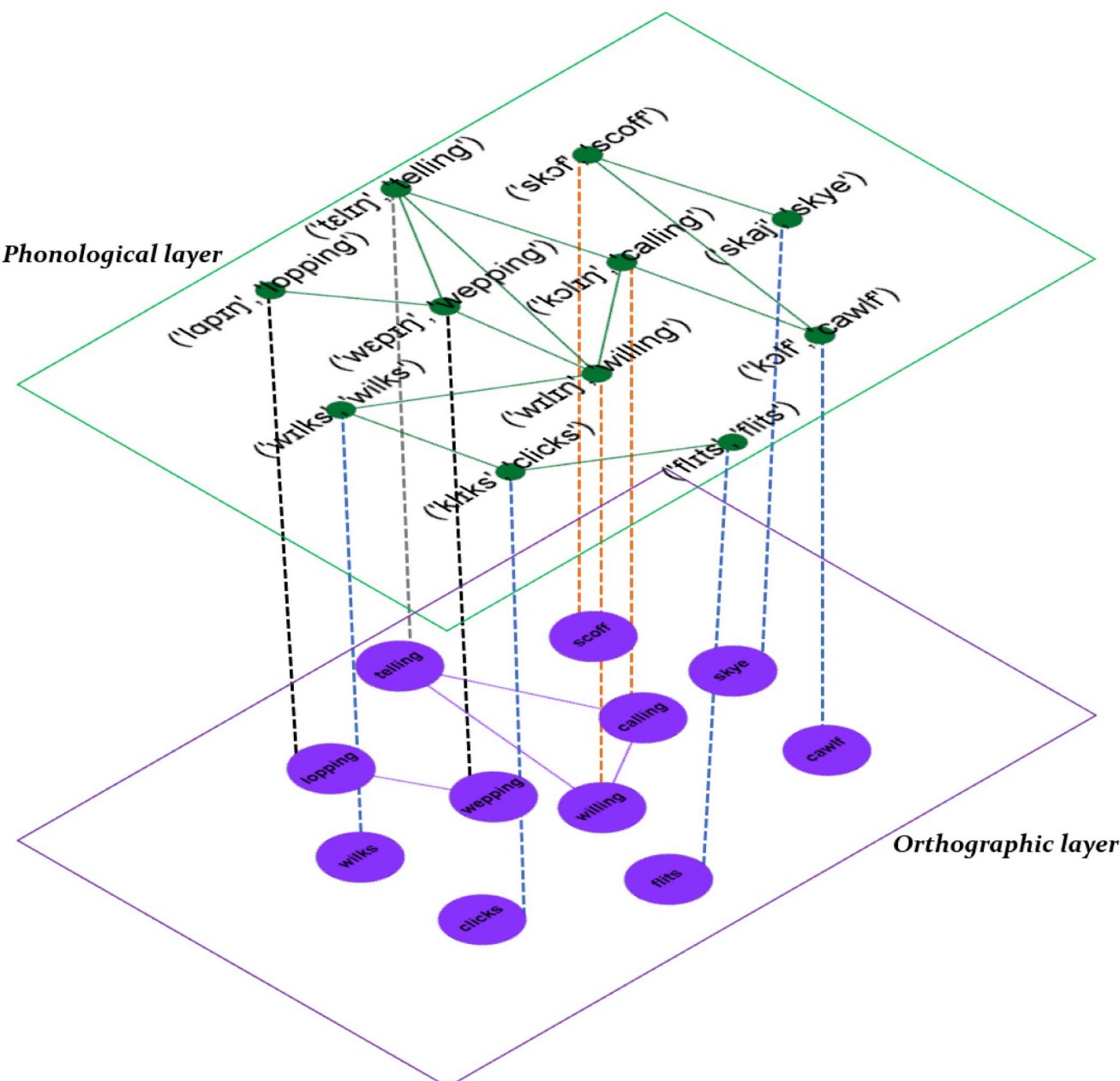

**Fig 1. Construction of the multiplex language network.** Schematic illustration of the construction of a multiplex language network for English based on an orthographic-distance and phonological-distance similarity networks. In the orthographic and phonological layers nodes are words and there is a link if the Damerau-Levenshtein distance is smaller than a given threshold ℓ. Notice that words in the phonological layer were translated into the International Phonetic Alphabet and then the DL was calculated.

multilayer networks have been studied in contexts such as finance and economics [17–19], social systems [20, 21], synchronization [22] and linguistics [12].

In this study, we plan to analyze the multiplex language network which consists of an orthographic network and phonological network (see Fig 1 for a schematic representation). For the orthographic network, we construct a network at word-level $G^{[O]} = (V^{[O]}, E^{[O]})$, where nodes are words and a link between two nodes is defined if the DL distance, described later, is smaller or equal than a threshold value ℓ. Similarly, a phonological

network $G^{[P]} = (V^{[P]}, E^{[P]})$ is constructed where the nodes represent words which were translated to the international phonological alphabet (IPA), and edges are defined if the DL, is smaller or equal than a given threshold $\ell$. To generate a multiplex language network at word-level, the orthographic and phonological networks are combined to form a two-layer word-level network, denoted by $G_L^{[\alpha]} = (V^{[\alpha]}, E^{[\alpha]})$, with $\alpha = O, P$. Here, the adjacency matrix for the multiplex network is given $a_{ij}^{[\alpha]}$, where $a_{ij}^{[\alpha]} = 1$ indicates that there is a link between node (word) $i$ and node (word) $j$ at layer $\alpha$. More formally, the adjacency matrix associated with each layer is defined as: $a_{ij}^{[\alpha]} = \Theta(\ell - d(w_i^{[\alpha]}, w_j^{[\alpha]})) - \delta_{ij}$, where $\Theta(-)$ represents the Heaviside function, $\delta_{ij}$ is the Kronecker delta and $d(w_i^{[\alpha]}, w_j^{[\alpha]})$ the DL distance between word $i$ and word $j$ at layer $\alpha$.

Regarding the distance condition between two words, as we mentioned in the Introduction, the distance similarity between two strings A and B can be defined as the minimum number of edit operations needed to transform A into B. These operations are: (1) substitute a character in A to a different character, (2) insert a character into A, (3) delete a character of A, and (4) transpose two adjacent characters of A. The Damerau-Levenshtein (DL) distance is then defined as the length of the optimal edit sequence. For instance, the Levenshtein distance is the length of the shortest sequence of substitutions, insertions, and deletions needed to transform string A into string B. In our analysis, we adopt the DL distance $\ell$ as a threshold value to define a link between two words.

## Databases

The corpus of words were constructed from written texts (books) freely available at Gutenberg project www.gutenberg.org. The written texts were pre-processed to remove function words, stop words and any mark symbol. The titles of the written texts and the resulting corpus are described in https://doi.org/10.6084/m9.figshare.12735380.v4 [23]. The final corpora contain $10^4$ words with their corresponding translation to the international phonetic alphabet for four languages (transliterated by the epitran library of Python version 3.6.8).

## Topological properties of single-layer and multiplex networks

Our initial analysis is focused on the basic topological characteristics of two individual networks, and then to proceed to investigate similarities and differences of the two layers. The single-layer-network measures (of a network with $N$ nodes) in a multiplex network that have been initially evaluated are [24]:

- Density. The density of a layer $\alpha$, $\rho^{[\alpha]}$, is given as:

$$\rho^{[\alpha]} = \frac{2m^{[\alpha]}}{N(N-1)} \tag{1}$$

where $m^{[\alpha]}$ is the number of actual connections within the layer $\alpha$.

- Degree distribution. The degree $k_i^{[\alpha]}$ of a node $i$ is the number of links outgoing (or incoming) to that node,

$$k_i^{[\alpha]} = \sum_{j=1}^{N} a_{ij}^{[\alpha]}. \tag{2}$$

The degree distribution for layer $\alpha$ is then defined as the fraction of nodes in the network

with degree $k$,

$$P^{[\alpha]}(k) = \frac{n_k^{[\alpha]}}{N}, \tag{3}$$

where $n_k^{[\alpha]}$ is the number of nodes with degree $k$.

- Clustering Coefficient. Measures the degree of transitivity in connectivity among the nearest neighbors of a node $i$ within the layer $\alpha$. $C_i^{[\alpha]}$ is calculated as [25],

$$C_i^{[\alpha]} = \frac{2E_i^{[\alpha]}}{k_i^{[\alpha]}(k_i^{[\alpha]} - 1)}, \tag{4}$$

where $E_i^{[\alpha]}$ is the number of links between the $k_i^{[\alpha]}$ neighbors of the node $i$ within the layer $\alpha$.

- Average Nearest-Neighbor Degree. Measures the average of the neighbors of a node [25]. The $\bar{k}_{nn,i}^{[\alpha]}$ is calculated as:

$$\bar{k}_{nn,i}^{[\alpha]} = \frac{1}{k_i^{[\alpha]}} \sum_{j=1}^{N} a_{ij}^{[\alpha]} k_j^{[\alpha]}. \tag{5}$$

- Modularity. Given $c_i^{[\alpha]}$ the community associated to the node $i$ within the layer $\alpha$, where $c_i^{[\alpha]} \in \{1, 2, \ldots, P\}$, with $P$ a natural number. The modularity, $Q^{[\alpha]}$ of a given layer $\alpha$ is given by [24]:

$$\mathcal{Q}^{[\alpha]} = \frac{1}{2m^{[\alpha]}} \sum_{ij} \left( a_{ij}^{[\alpha]} - \frac{k_i^{[\alpha]} k_j^{[\alpha]}}{2m^{[\alpha]}} \right) \delta(c_i^{[\alpha]} c_j^{[\alpha]}), \tag{6}$$

where $\delta$ is the Kronecker delta. We use the Louvain algorithm [26] to perform a greedy optimization of the modularity.

In order to get insight on our study, we plan to characterize structural network properties and information-based quantities from the following perspectives: (i) presence of similarities across 4 linguistics families (Romance, Germanic, Slavic and Uralic), (ii) increase of the size of the number of words (corpus) from $10^4$ to $50 \times 10^3$, and (iii) robustness of the networks. Regarding (i), we will analyze to what extent the topological single-layer and multiplex network properties exhibit similarities and differences quantified by means of correlation measures and information-theory-based metrics for 12 natural languages which belong to 4 linguistic families. To reinforce the characterization of the grouping patterns of nodes of the network, we will consider multilayer community detection algorithms [27] to determine the presence of clusters across layers. These procedures will help us in the understanding of local and global network properties of the orthographic-phonological variations across several languages. With respect to (ii), we plan to increase the size of the corpus to $50 \times 10^3$ in the number words for all languages in our study. The results for this size will confirm the validity of our preliminary results for $10^4$ words, and also will permit to evaluate the concordance of our findings with previous results. Concerning (iii), the robustness of the single-layer and multiplex network will be evaluated by means of two well-recognized strategies: random removal of fraction of nodes and edges and directed attacks [28]. Moreover, a randomized version of the networks will be also considered to repeat all the calculations in our study.

## Initial analyses for 4 natural languages and $10^4$ words

We have started our analyses working with 4 languages (Spanish, English, German and Russian) with corpus containing $10^4$ words each one. Table 1 concentrates the results of the calculations for the basic structural properties of the orthographic network, the phonological network and the multiplex one. These preliminary results of topological features indicate that there are common properties at local and global scales. Interestingly, the results for the average clustering for Spanish, in the case of the phonological layer with $\ell = 2$, is concordant with the value reported for phonological networks [16], where the authors used a different corpus and

**Table 1. The basic topological network quantities are listed for the ortographic ($G^O$) and phonological ($G^P$) networks.**

| Language | Metric | | $G^O$ | | | $G^P$ | | |
|---|---|---|---|---|---|---|---|---|
| | Threshold | $\ell = 1$ | $\ell = 2$ | $\ell = 3$ | $\ell = 1$ | $\ell = 2$ | $\ell = 3$ |
| English | Density | | $0.60(10^{-3})$ | $1.98(10^{-3})$ | $10.23(10^{-3})$ | $0.86(10^{-3})$ | $3.18(10^{-3})$ | $15.42(10^{-3})$ |
| | Average degree $\bar{k}$ | | 2.03 | 13.74 | 91.53 | 1.24 | 3.41 | 16.78 |
| | Clustering $\bar{c}$ | | 0.12 | 0.22 | 0.29 | 0.03 | 0.12 | 0.21 |
| | Nearest neighbor $\bar{k}_{nn}$ | | 2.51 | 19.31 | 137.66 | 0.46 | 4.57 | 24.92 |
| | Maximum modularity $Q$ | | 0.91 | 0.55 | 0.35 | 0.99 | 0.77 | 0.49 |
| | Fit exponent $\gamma$ | | $2.57 \pm 0.57$ | $1.31 \pm 0.25$ | $1.02 \pm 0.16$ | $2.08 \pm 0.56$ | $1.22 \pm 0.24$ | $0.96 \pm 0.18$ |
| | Average cluster size | | 4.26 | 13.76 | 58.10 | 5.24 | 17.35 | 60.07 |
| German | Density | | $0.59(10^{-3})$ | $1.05(10^{-3})$ | $4.29(10^{-3})$ | $0.81(10^{-3})$ | $1.61(10^{-3})$ | $5.70(10^{-3})$ |
| | Average degree $\bar{k}$ | | 1.47 | 5.73 | 32.23 | 1.93 | 8.53 | 42.24 |
| | Clustering $\bar{c}$ | | 0.10 | 0.22 | 0.27 | 0.14 | 0.22 | 0.29 |
| | Nearest neighbor $\bar{k}_{nn}$ | | 1.72 | 7.77 | 48.76 | 2.35 | 11.36 | 61.48 |
| | Maximum modularity $Q$ | | 0.98 | 0.67 | 0.45 | 0.94 | 0.64 | 0.47 |
| | Fit exponent $\gamma$ | | $3.14 \pm 0.48$ | $1.82 \pm 0.42$ | $1.20 \pm 0.21$ | $2.47 \pm 0.47$ | $1.57 \pm 0.37$ | $1.15 \pm 0.12$ |
| | Average cluster size | | 2.94 | 8.05 | 21.83 | 3.68 | 8.56 | 23.49 |
| Russian | Density | | $0.64(10^{-3})$ | $0.65(10^{-3})$ | $2.68(10^{-3})$ | $0.74(10^{-3})$ | $0.65(10^{-3})$ | $2.16(10^{-3})$ |
| | Average degree $\bar{k}$ | | 1.22 | 3.73 | 22.27 | 1.24 | 3.42 | 16.78 |
| | Clustering $\bar{c}$ | | 0.06 | 0.19 | 0.25 | 0.07 | 0.19 | 0.25 |
| | Nearest neighbor $\bar{k}_{nn}$ | | 1.35 | 5.06 | 33.92 | 1.37 | 4.57 | 24.92 |
| | Maximum modularity $Q$ | | 0.99 | 0.74 | 0.43 | 0.99 | 0.78 | 0.5 |
| | Fit exponent $\gamma$ | | $4.17 \pm 0.50$ | $2.09 \pm 0.35$ | $1.31 \pm 0.25$ | $3.64 \pm 0.34$ | $2 \pm 0.26$ | $1.39 \pm 0.22$ |
| | Average cluste size | | 2.40 | 6.18 | 25.52 | 2.40 | 5.24 | 17.31 |
| Spanish | Density | | $0.52(10^{-3})$ | $0.86(10^{-3})$ | $4.14(10^{-3})$ | $0.52(10^{-3})$ | $1.11(10^{-3})$ | $5.59(10^{-3})$ |
| | Average degree $\bar{k}$ | | 1.41 | 5.87 | 37.36 | 1.68 | 8.07 | 51.53 |
| | Clustering $\bar{c}$ | | 0.06 | 0.21 | 0.27 | 0.09 | 0.23 | 0.28 |
| | Nearest neighbor $\bar{k}_{nn}$ | | 1.66 | 8.18 | 56.69 | 2.01 | 11.28 | 77.20 |
| | Maximum modularity $Q$ | | 0.98 | 0.65 | 0.41 | 0.95 | 0.59 | 0.39 |
| | Fit exponent $\gamma$ | | $3.26 \pm 0.53$ | $1.90 \pm 0.46$ | $1.19 \pm 0.31$ | $2.98 \pm 0.49$ | $1.77 \pm 0.43$ | $1.12 \pm 0.30$ |
| | Average cluster size | | 2.87 | 10.21 | 60.49 | 3.36 | 12.72 | 83.75 |

Notes. Topological metrics of the orthographic network and the phonological network. Here we present the average values of the degree ($k_i$), clustering ($c_i$) and nearest neighbor ($k_{nn,i}$). We observe that the density, $\bar{k}$, $\bar{c}$ and $\bar{k}_{nn}$ exhibit an increasing behavior for the four languages and the two layers, with some similarities such as it occurs for $\bar{c}$ in both layers and distances $\ell = 2, 3$. For the modularity and the average cluster size, we observe they exhibit opposite trends, while the modularity decreases as $\ell$ increases, the average cluster size increases because a larger number of nodes tends to be connected to a giant component.

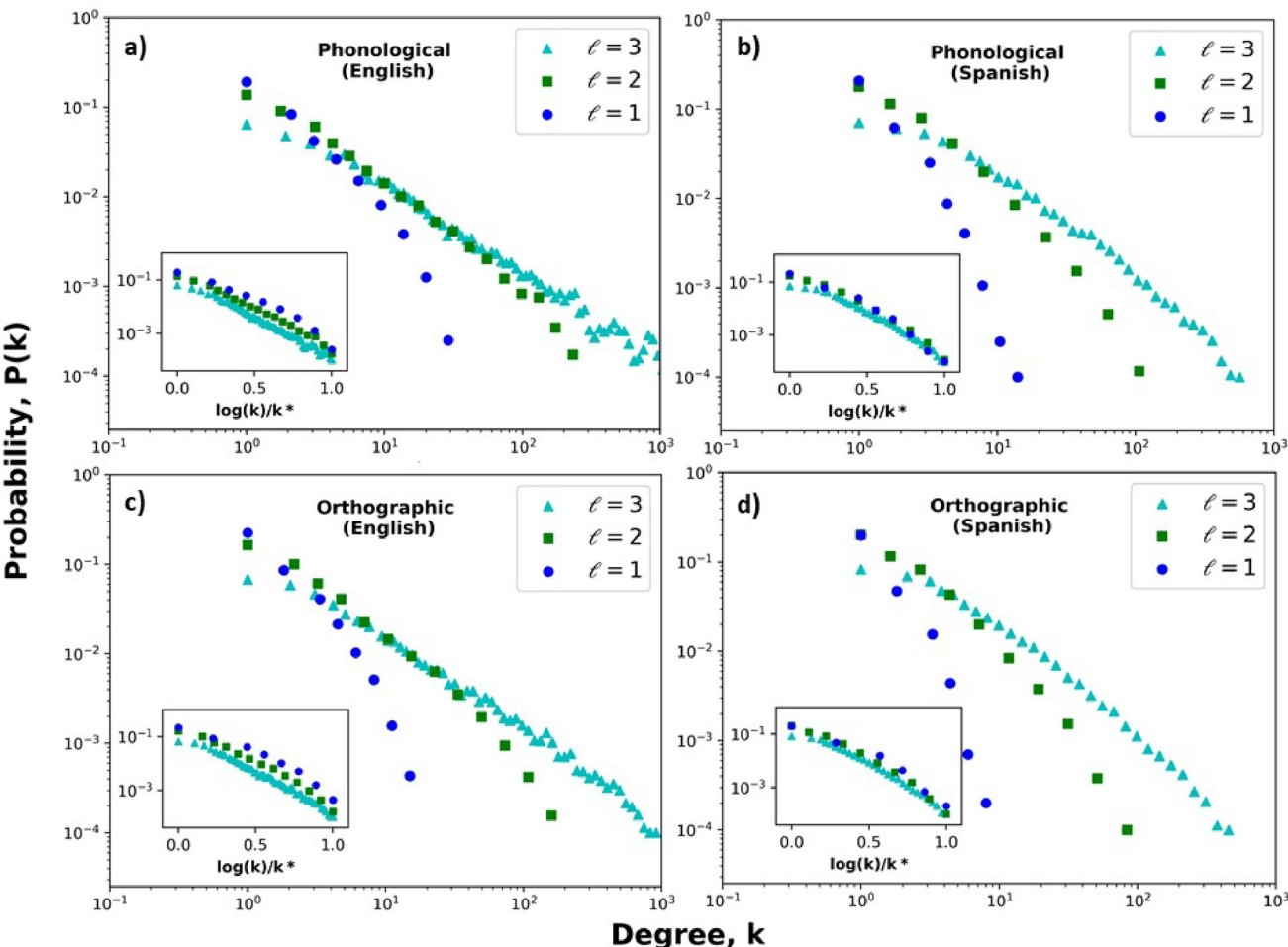

**Fig 2. Degree distributions for phonological and orthographic networks and several DL distance thresholds.** a) Phonological (English). b) Phonological (Spanish). c) Orthographic (English). d) Orthographic (Spanish). For a better comparison of the data, the insets of each plot show the corresponding degree distribution for normalized degrees, where $k^* = \max(\log(k))$.

assumed an edge between words if the differ by a single phoneme or sound segment. In order to get a better understanding of the patterns of the connectivities in both layers, we proceed to construct the degree distribution for different threshold values of the DL distance ranging form 1 to 3. Fig 2 shows the cases of the degree distributions of $G^O$ and $G^P$ for Spanish, German, English and Russian and DL distances from 1 to 3. It is visually apparent that, for the 4 languages, as the DL distance increases, the distributions change from an approximately exponential regime ($\ell = 1$) to a combination of an exponential and power law behavior ($\ell \geq 2$). It is likely that the best fit would be obtained by means of a truncated power law function, which has been suggested to fit phonological networks [16]. In our initial estimation of the best fit of the distributions, we only consider the power law behavior at the tails, $P(k) \sim k^{-\gamma}$, where $\gamma$ is an exponent which characterizes the connectivities. For instance, for a DL distance $\ell = 3$ and the phonological layer, the estimated $\gamma$-exponents (1.12) for the power law degree distribution is concordant to the value reported in [16] for Spanish. Additional tests are needed in order to get a better description of the distributions, and also for the behavior of the other topological metrics as a function of the degree.

## Proposed timeline

The proposed study requires at most 3 months to complete (starting Dec. 1st., 2020). It is planned to build the corpus of 12 new languages and enlarge the size of the existing ones to $50 \times 10^3$. This stage is planned to conclude in a month, and immediately proceed to carry out the corresponding pre-processing for the translation into the international phonetic alphabet of all the corpus. Then we will proceed with the calculations of the metrics of the orthographic, phonological and multiplex networks. Next, we plan on finishing data interpretation and drafting the final manuscript in the following two months.

## Supporting information

**S1 File.**
(TXT)

## Author Contributions

**Conceptualization:** Bibiana Obregón-Quintana, Lev Guzmán-Vargas.

**Data curation:** Pablo Lara-Martínez, Cesar F. Reyes-Manzano, Irene López-Rodríguez.

**Formal analysis:** Lev Guzmán-Vargas.

**Investigation:** Bibiana Obregón-Quintana, Lev Guzmán-Vargas.

**Methodology:** Bibiana Obregón-Quintana, Cesar F. Reyes-Manzano, Irene López-Rodríguez.

**Software:** Pablo Lara-Martínez.

**Validation:** Pablo Lara-Martínez.

**Writing – original draft:** Lev Guzmán-Vargas.

**Writing – review & editing:** Bibiana Obregón-Quintana, Lev Guzmán-Vargas.

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
