## [Decision Letter · Decision Letter 0]

11 Nov 2020

PONE-D-20-23977

Comparing phonological and orthographic networks: a multiplex analysis

PLOS ONE

Dear Dr. Guzmán-Vargas,

Thank you for submitting your manuscript to PLOS ONE. After careful consideration, we feel that it has merit but does not fully meet PLOS ONE’s publication criteria as it currently stands. Therefore, we invite you to submit a revised version of the manuscript that addresses the points raised during the review process. Please notice that only comments that are applicable to a Registered Report Protocol should be addressed. You can consider additional suggestions when preparing a Registered Report.

We look forward to receiving your revised manuscript.

Kind regards,

Diego Raphael Amancio

Academic Editor

PLOS ONE

Journal Requirements:

"This work was partially supported by COFAA-IPN, EDI-IPN, and Conacyt-Mexico.".

i) We note that you have provided funding information that is not currently declared in your Funding Statement. However, funding information should not appear in the Acknowledgments section or other areas of your manuscript. We will only publish funding information present in the Funding Statement section of the online submission form.

ii) Please remove any funding-related text from the manuscript and let us know how you would like to update your Funding Statement. Currently, your Funding Statement reads as follows:

 "NO".

 iii) Please include your amended statements within your cover letter; we will change the online submission form on your behalf.

"NO".

i) Please complete your Competing Interests on the online submission form to state any Competing Interests. If you have no competing interests, please state "The authors have declared that no competing interests exist.", as detailed online in our guide for authors at http://journals.plos.org/plosone/s/submit-now

 ii) This information should be included in your cover letter; we will change the online submission form on your behalf.

Additional Editor Comments (if provided):

The authors should address only the comments that are applicable for a Registered Report Protocol.

Reviewers' comments:

Reviewer's Responses to Questions

**Comments to the Author**

1. Does the manuscript provide a valid rationale for the proposed study, with clearly identified and justified research questions?

Reviewer #1: Yes

Reviewer #2: Yes

2. Is the protocol technically sound and planned in a manner that will lead to a meaningful outcome and allow testing the stated hypotheses?

Reviewer #1: Yes

Reviewer #2: Yes

3. Is the methodology feasible and described in sufficient detail to allow the work to be replicable?

Reviewer #1: Yes

Reviewer #2: Yes

4. Have the authors described where all data underlying the findings will be made available when the study is complete?

Reviewer #1: Yes

Reviewer #2: Yes

5. Is the manuscript presented in an intelligible fashion and written in standard English?

Reviewer #1: Yes

Reviewer #2: Yes

6. Review Comments to the Author

You may also provide optional suggestions and comments to authors that they might find helpful in planning their study.

Reviewer #1: The study proposed by the authors in this report is interesting and could yield interesting results. The authors do a good job in motivating their project. The manuscript is well written barring some minor typos (see below).

The authors may consider adopting some multilayer community detection algorithms like the ones described in [1] for example to find more interesting clusters across layers.

Typo on line 79 on page 3 - A & B should be V & W.

Reference:

[1] De Bacco, Caterina, et al. "Community detection, link prediction, and layer interdependence in multilayer networks." Physical Review E 95.4 (2017): 042317.

Reviewer #2: The manuscript ‘Comparing phonological and orthographic networks: a multiplex analysis’ by Pablo Lara-Martinez, et al. presents several graph-theoretic statistics and metrics of networks build from orthographic and phonological corpora of words for four languages (English, German Russian, and Spanish). The authors assign words to nodes and use the Damerau-Levenshtein distance to create edges between nodes depending on their distance. Several networks are created in which the cutoff for the distance is different and the statistics are listed for each network. The authors also propose a timeline to expand the size of the corpora and repeat the experiments, but this part of the analysis is not included in the manuscript.

The main issue with this manuscript is that it does not include a discussion or interpretation of the results. Even though the methods are described and results are listed, there are no scientific claims, there is no analysis of the results, there is no real comparison to relevant work, and there is no discussion of the importance of the results or their significance to the field.

The manuscript is unsittable for publication as a scientific article in the present form but could be published if a discussion section is included.

I also have minor suggestions:

• Make the networks created with the different distance cutoffs publicly available. Calculating the DL distance is not necessarily trivial and this dataset would be useful to other researchers, who can use it to compare their work or methods to the current ones.

• In the text, define DL as ‘Damerau-Levenshtein’ before using the acronym.

• I did not identify any connection in the results listed between the phonological and the orthographic networks other than 1 to 1, so I don’t think this is rigorously speaking multiplex.

• The caption of Fig. 1 states that the illustration is for Spanish but the words are in English in the actual figure.

• Although it is understandable, V should be defined for completeness before Eq. 3.

• M is used in the definition of the degree of a node right before Eq. 3, but M is never defined. From inspection it seems like M is the multiplex layer, but this should be stated clearly to make it easier for the reader.

• I did not understand the definition of the modularity (Eq. 6). It says that the partition includes elements C_1, …, C_M. In this case M is subscript. C is defined as the clustering coefficient in Eq. 4. The set S is not really defined. So this needs more work.

• The results in Fig. 2 might not be apples to apples because the highest degree depends on the DL distance cutoff. I suggest normalizing the horizontal axis by the highest degree in each network.

• I suggest removing Fig. 3 and mentions of specific start dates and due dates. This is not really part of a scientific paper.

7. PLOS authors have the option to publish the peer review history of their article (what does this mean?). If published, this will include your full peer review and any attached files.

Reviewer #1: No

Reviewer #2: No

---

## [Author Response · Author response to Decision Letter 0]

25 Nov 2020

Reviewer #1

“The study proposed by the authors in this report is interesting and could yield interesting results. The authors do a good job in motivating their project. The manuscript is well written barring some minor typos (see below). The authors may consider adopting some multilayer community detection algorithms like the ones described in [1] for example to find more interesting clusters across layers.

Typo on line 79 on page 3 - A & B should be V & W.

Reference:

[1] De Bacco, Caterina, et al. "Community detection, link prediction, and layer interdependence in multilayer networks." Physical Review E 95.4 (2017): 042317.”

Response: Thank you for your comments. We have added the application of the suggested multilayer community dectection method to the revised version of our Protocol (Lines 129-131), which will enrich our discussion about potential grouping of nodes (words) in the context of the two-layer network. The typo was corrected. 

Reviewer #2

1)• Make the networks created with the different distance cutoffs publicly available. Calculating the DL distance is not necessarily trivial and this dataset would be useful to other researchers, who can use it to compare their work or methods to the current ones.

Response: Thank you for your suggestion. We have added the corresponding networks (with 10⁴ nodes) for each corpus to our dataset in the Figshare repository. We plan to update the dataset once we have concluded our study. 

2) • In the text, define DL as ‘Damerau-Levenshtein’ before using the acronym.

Response: Corrected in the revised version (Line 23)

3)• I did not identify any connection in the results listed between the phonological and the orthographic networks other than 1 to 1, so I don’t think this is rigorously speaking multiplex.

Response: We plan to address the multiplex features in the extended paper 

4)• The caption of Fig. 1 states that the illustration is for Spanish but the words are in English in the actual figure.

Response: The error was corrected

5)• Although it is understandable, V should be defined for completeness before Eq. 3.

Response: We have clarified the network notation, and the meaning of the letters (see lines 99-121 )

6)• M is used in the definition of the degree of a node right before Eq. 3, but M is never defined. From inspection it seems like M is the multiplex layer, but this should be stated clearly to make it easier for the reader.

Response: We have corrected the description of the network’s notation (see lines 99-121)

7)• I did not understand the definition of the modularity (Eq. 6). It says that the partition includes elements C_1, …, C_M. In this case M is subscript. C is defined as the clustering coefficient in Eq. 4. The set S is not really defined. So this needs more work.

Response: Thank you for your comment. In the revised version of the manuscript we have clarified our notation regarding the modularity definition (see Lines 117-121).

8)• The results in Fig. 2 might not be apples to apples because the highest degree depends on the DL distance cutoff. I suggest normalizing the horizontal axis by the highest degree in each network.

Response: We have added the suggested normalization of the horizontal axis in the new version of Fig. 2

9)• I suggest removing Fig. 3 and mentions of specific start dates and due dates. This is not really part of a scientific paper.

Response: The mentioned fig was removed. The new dates were updated in the main text.

---

## [Decision Letter · Decision Letter 1]

28 Dec 2020

Comparing phonological and orthographic networks: a multiplex analysis

PONE-D-20-23977R1

Dear Dr. Guzmán-Vargas,

We’re pleased to inform you that your manuscript has been judged scientifically suitable for publication and will be formally accepted for publication once it meets all outstanding technical requirements.

Kind regards,

Diego Raphael Amancio

Academic Editor

PLOS ONE

Additional Editor Comments (optional):

Reviewers' comments:

Reviewer's Responses to Questions

**Comments to the Author**

1. Does the manuscript provide a valid rationale for the proposed study, with clearly identified and justified research questions?

Reviewer #2: Yes

2. Is the protocol technically sound and planned in a manner that will lead to a meaningful outcome and allow testing the stated hypotheses?

Reviewer #2: Yes

3. Is the methodology feasible and described in sufficient detail to allow the work to be replicable?

Reviewer #2: Yes

4. Have the authors described where all data underlying the findings will be made available when the study is complete?

Reviewer #2: Yes

5. Is the manuscript presented in an intelligible fashion and written in standard English?

Reviewer #2: Yes

6. Review Comments to the Author

You may also provide optional suggestions and comments to authors that they might find helpful in planning their study.

Reviewer #2: The authors satisfactorily addressed in their revised manuscript the points raised in the first round of reviews and I recommend publication of the manuscript.

7. PLOS authors have the option to publish the peer review history of their article (what does this mean?). If published, this will include your full peer review and any attached files.

Reviewer #2: No

---

## [Editor Report · Acceptance letter]

7 Jan 2021

PONE-D-20-23977R1 

Comparing phonological and orthographic networks: a multiplex analysis 

Dear Dr. Guzmán-Vargas:

I'm pleased to inform you that your manuscript has been deemed suitable for publication in PLOS ONE. Congratulations! Your manuscript is now with our production department. 

Kind regards, 

on behalf of

Dr. Diego Raphael Amancio 

Academic Editor

PLOS ONE